# Audit and feedback to change diagnostic image ordering practices: A systematic review and meta-analysis

**Oluwatosin Badejo**[1], **Maria Saleeb**[1], **Amanda Hall**[1,2], **Bradley Furlong**[1], **Gabrielle S. Logan**[1], **Zhiwei Gao**[2], **Brendan Barrett**[2,3], **Lindsay Alcock**[4], **Kris Aubrey-Bassler**[1,2]*

1 Primary Healthcare Research Unit, Faculty of Medicine, Memorial University of Newfoundland and Labrador, St. John's, Newfoundland and Labrador, Canada, 2 Population Health and Applied Health Sciences, Faculty of Medicine, Memorial University of Newfoundland and Labrador, St. John's, Newfoundland and Labrador, Canada, 3 Discipline of Medicine, Faculty of Medicine, Memorial University of Newfoundland, Newfoundland and Labrador, St. John's, Newfoundland and Labrador, Canada, 4 Health Sciences Library, Memorial University of Newfoundland and Labrador, Newfoundland and Labrador, St. John's, Newfoundland and Labrador, Canada

* kaubrey@mun.ca

**Data Availability Statement:** All relevant data are within the manuscript and its Supporting information files.

## Abstract

### Background

Up to 30% of diagnostic imaging (DI) tests may be unnecessary, leading to increased healthcare costs and the possibility of patient harm. The primary objective of this systematic review was to assess the effect of audit and feedback (AF) interventions directed at healthcare providers on reducing image ordering. The secondary objective was to examine the effect of AF on the appropriateness of DI ordering.

### Methods

Studies were identified using MEDLINE, EMBASE, CINAHL, Cochrane Central Register of Controlled Trials and ClinicalTrials.gov registry on December 22nd, 2022. Studies were included if they were randomized control trials (RCTs), targeted healthcare professionals, and studied AF as the sole intervention or as the core component of a multi-faceted intervention. Risk of bias for each study was evaluated using the Cochrane risk of bias tool. Meta-analyses were completed using RevMan software and results were displayed in forest plots.

### Results

Eleven RCTs enrolling 4311 clinicians or practices were included. AF interventions resulted in 1.5 fewer image test orders per 1000 patients seen than control interventions (95% confidence interval (CI) for the difference -2.6 to -0.4, p-value = 0.009). The effect of AF on appropriateness was not statistically significant, with a 3.2% (95% CI -1.5 to 7.7%, p-value = 0.18) greater likelihood of test orders being considered appropriate with AF vs control interventions. The strength of evidence was rated as moderate for the primary objective but was very low for the appropriateness outcome because of risk of bias, inconsistency in findings, indirectness, and imprecision.

**Funding:** O.B received the Memorial University of Newfoundland's NL SUPPORT – Support for People and Patient Oriented Research and Trials Grant (https://www.mun.ca/nlcahr/nlcahr-funding-programs/nl-support/nl-support-patient-oriented-research-grant/). There is no grant number associated with this award. The funder played no role in the study design, data collection and analysis, decision to publish, or preparation of the manuscript.

**Competing interests:** No authors have competing interests.

## Conclusion

AF interventions are associated with a modest reduction in total DI ordering with moderate certainty, suggesting some benefit of AF. Individual studies document effects of AF on image order appropriateness ranging from a non-significant trend toward worsening to a highly significant improvement, but the weighted average effect size from the meta-analysis is not statistically significant with very low certainty.

## Introduction

Up to thirty percent of diagnostic imaging (DI) tests may be unnecessary [1, 2] and this excess use increases healthcare costs, wait times, and the likelihood of patient harm [3]. Unwarranted DI testing often leads to incidental findings which can in turn lead to a cascade of further unnecessary tests and treatments [4, 5]. For example, more liberal use of imaging for back pain has been associated with higher rates of surgery and other procedures and higher healthcare costs, as well as longer absence from work [6]. Incidental findings can lead to increased patient anxiety, financial burden, and ultimately delays in necessary treatment [5, 7], while also exacerbating long wait times for patients who do require these tests [8]. Physical harm to patients is also important to consider, as some types of imaging such as computed tomography (CT) involve exposure to high doses of ionizing radiation, which may lead to an increased risk of iatrogenic cancers [9, 10].

Audit and feedback (AF) has been implemented in healthcare settings as a strategy to modify behaviours in delivering health care services, including DI test ordering [11]. AF provides summaries of clinical performance over a specified period to health care providers, with the aim of motivating behaviour change. A Cochrane review of 70 randomized trials in healthcare settings revealed moderate quality evidence that AF has a moderate effect (dichotomous outcome: median adjusted risk difference of 4.3%, IQR 0.5% to 16% (49 studies); continuous outcome: median adjusted percent change of 1.3% (interquartile range (IQR) 1.3% to 28.9% (26 studies)) on increasing health professional compliance with desired behaviour when compared to usual practice [11]. This review included 4363 providers or provider groups from 49 trials that examined dichotomous outcomes and 1266 providers or provider groups from 21 trials that examined continuous outcomes. However, this review examined AF that targeted multiple issues, including the management of diabetes mellitus, blood pressure control, inappropriate antibiotic prescribing, X-ray utilization rates and more. The large range of topics addressed in this review and the heterogeneity in outcomes make it difficult to draw conclusions about the effect of AF on DI ordering, and the effect estimate for this area was not provided [11].

The objective of the current review was to determine the effect of AF interventions on DI ordering rates and DI ordering appropriateness. We also completed a comprehensive description of DI AF interventions using the template for intervention description and replication (TIDieR) checklist [12].

## Materials and methods

Our review protocol was developed in line with recommendations from the Cochrane Effective Practice and Organization of Care (EPOC) group [13] and was prospectively registered with Open Science Framework (https://osf.io/5dczr) [14]. Although this group is no longer active, the resources are still published online [13] and additional information can be found in

the Cochrane handbook [15]. Initially, we proposed to include both randomized controlled trials (RCTs) and some observational designs; however, our literature search discovered a sufficient number of RCTs and a discrepancy in results between the RCTs and the observational studies. The primary analyses were therefore limited to RCTs due to higher quality evidence, and the findings of the observational studies are reported in the S1 Appendix. We also proposed to compare AF interventions to a different active intervention, but elected to remove this comparison to simplify interpretation of the findings. Finally, the original protocol included only the appropriateness of image orders as the sole outcome, but we elected to add a total DI orders outcome because it directly aligned with the purpose of this review.

## Data sources

We identified studies using a systematic search of MEDLINE (PubMed), EMBASE, CINAHL, the Cochrane Central Register of Controlled Trials and the ClinicalTrials.gov registry. Our search strategy was modelled after that of the Cochrane review [11], but it was adapted by an information specialist to ensure it included sufficient terms related to diagnostic imaging (S1 Appendix). These search strategies also underwent peer review using the Peer Review of Electronic Search Strategy (PRESS) guidelines [16]. We searched for full-text articles available up to December 20th, 2022 with no earlier date restriction. These database searches were supplemented with electronic and manual searches, including forward tracking to identify papers that cited the studies already included in the review.

## Inclusion and exclusion criteria

**Study design.**   RCTs with no restriction on language, geographic setting or year of publication were included in the primary analyses. We planned to translate articles published in a language other than English using Google Translate (https://translate.google.com/) which is as accurate as human translators for the languages commonly used for science [17]. The results of controlled before-after, non-randomized controlled studies and interrupted time series analyses are included in the S1 Appendix. Uncontrolled studies, case series and case reports were excluded.

**Population.**   We included studies targeting health-care professionals who order DI in the routine management of their patients. Studies were excluded if the target population was healthcare professionals who do not normally order imaging tests such as pharmacists, radiologists, technicians, and medical students.

**Intervention and comparator.**   Studies that provided feedback on individual clinician or clinician group ordering compared to a target recommended by local, regional or national guidelines, or a benchmark such as test ordering among peer clinicians were included. Studies that examined AF as the sole strategy or AF as the integral part of a multi-faceted intervention were included. Similar to the Cochrane review of AF [11], we considered AF to be "integral" if the other features of the intervention were unlikely to be offered without AF or if other components of the intervention were optional and therefore not necessarily received or used by subjects in the intervention group. For example, we included studies of AF combined with an educational session, but we excluded a comparison of AF combined with an electronic, point-of-care decision support tool vs usual practice from this sub-analysis.

We were primarily interested in the comparison of AF to a usual practice control group. We did not consider the provision of paper or digital clinical practice guidelines to be an active intervention as provision of guidelines alone is rarely associated with measurable behaviour change [18]. Thus, groups receiving guidelines together with AF were categorized as "AF alone," and those receiving guidelines alone were categorized as "usual practice." If

comparison groups were not explicitly defined, they were assumed to be usual practice. Some papers studied AF combined with another intervention verses the other intervention alone, which we included in a separate subgroup of the meta-analyses as the effect of adding AF to another intervention may be different than the effect of AF alone.

Studies were excluded if audits occurred without feedback, if they occurred during a patient visit, or if feedback was given in real time during or shortly after a patient encounter. If feedback was given for hypothetical situations or was a reminder without reference to specific ordering behaviour, it was also excluded. We were also exclusively interested in the effect of AF on *diagnostic* imaging and we therefore excluded studies that focussed on *screening* tests such as mammograms.

**Outcome.** The primary outcome of this review was the total number of DI tests ordered and the secondary outcome was the appropriateness of test orders. Appropriateness was measured as the total number or proportion of imaging tests that were classified as concordant with a standard of care, such as clinical practice guidelines, according to the individual study authors. Some papers examined AF of non-DI tests in addition to DI orders, but only the DI-specific outcome data were included.

When studies reported more than one measure of the same outcome, we extracted (in order of preference): post-intervention continuous measure adjusted for baseline values, change from baseline continuous measure, post-intervention continuous measure (no adjustment for baseline values), then post-intervention dichotomous measure. Odds ratios for dichotomous outcomes were converted to continuous outcomes as described in the Cochrane Handbook [15, Section 10.6] to be included in the continuous meta-analyses.

## Study selection

We uploaded the identified citations to the web-based systematic review software platform, Covidence [19]. Duplicates were identified automatically by Covidence or manually during screening. The titles and abstracts of all articles were screened independently by two review authors and screening conflicts were resolved by a third reviewer (OB, MS, AH, KAB). Pilot screening of 10 studies was undertaken to ensure uniformity in screening procedure. Full-text screening followed the same process of review and conflict resolution.

## Data extraction and quality assessment

We extracted information on study characteristics, population, intervention and outcome from each of the included studies according to the TIDieR recommendations [12]. We also extracted all data relating to the outcomes described above including raw numbers, proportions and effect estimates where provided using a modified version of the Cochrane Effective Practice and Organization of Care (EPOC) data collection checklist [20]. Data were extracted independently by two reviewers and discrepancies in the extracted data were resolved by discussion or involvement of a third reviewer (OB, MS, AH, KAB).

The risk of bias for each included study was independently reviewed by two authors as high, low, or unclear using the Cochrane risk of bias tool version 1 (OB, BF, MS, KAB) [21], which has since been updated [22, 23] but was the version in use at the time this study was originally conceptualized. Studies with high risk of bias in at least one of the domains for assessment received an overall judgment of high risk. Studies with baseline imbalances in study group characteristics that were greater than would be expected due to chance were classified as high risk. Blinding study participants to AF interventions is not possible and the primary outcome was objective, so studies were not classified as high risk in this domain if investigators treated all study groups equally (e.g. both intervention and comparator groups

were aware they were participating in a study). In addition, because our primary outcome was objective, the unavailability of a pre-published protocol was not considered an indicator of selective reporting. Discrepancies in ratings were resolved through discussion or by involvement of a third reviewer.

## Data synthesis and analysis

All studies identified the individual clinician or clinical team as the unit of study participation. While some studies reported outcomes at the clinician or team level, some studies only reported results at the patient level (e.g., proportion of patient visits at which a DI test was ordered) or at the study group level (e.g., total DI orders in the group), without mentioning any adjustment for clustering of observations. Others reported results at the study participant level (e.g., mean number of DI orders per clinician), and other studies reported both. Although effect estimates measured at the study group or individual patient level are representative, variance is likely underestimated unless the analyses adjust for correlated observations. Therefore, we preferentially extracted data at the participating clinician level. We included data that did not appear to be adjusted for clustering but noted this when evaluating risk of bias and in our results.

Where possible, the means of multiple outcomes from the same paper (e.g. DI ordering for different imaging types) were included in the meta-analyses as recommended in the Cochrane Handbook [15, Section 6.5.2.10]. When it was not possible to determine the mean of outcomes (e.g., only odds ratios and 95% CIs reported), we included data for the more frequent outcome. Data were compiled into meta-analyses and forest plots, and heterogeneity was estimated using $I^2$ and $Chi^2$ statistics using Review Manager (RevMan) software [24]. Potential sources of heterogeneity are explored qualitatively in Results and Discussion. We considered presenting data in subgroups by imaging modality or by target organ, but there are few studies and heterogeneity exists primarily within potential subgroups rather than between subgroups so we elected to present data without such grouping. Because of variability in the continuous outcome measures used between studies, we combined results using the standardized mean difference (SMD). We then rescaled the summary SMD from the total imaging orders meta-analysis into units of the mean difference between intervention and control in the number of DI tests ordered per 1000 patient consultations, which was the outcome used across a plurality of studies in the meta-analysis, including the largest. This conversion of SMD to natural units is recommended by the Cochrane Collaboration to enhance interpretability of the SMD [25, 26]. SMD measures outcomes in units of the standard deviation; therefore, to convert the summary SMD and its confidence interval (CI) into natural units, we chose the weighted (by trial sample size) average of the SDs from each of the studies that used the DI tests per 1000 patients outcome. We also expressed this outcome as a percentage of the weighted average of the baseline, pre-intervention DI tests per 1000 patients from each of these same studies. The SMDs from the secondary outcome meta-analysis were similarly converted into the difference between intervention and control in the percentage of image test orders that were considered to be appropriate.

## Summary of findings and GRADE strength of evidence

Two authors (OB, MS), with resolution of disagreement by a third author (KAB), applied the Grading of Recommendations Assessment, Development and Evaluation (GRADE) approach to summarize our findings and rate the strength of evidence [15, Chapter 14]. The GRADE approach uses risk of bias, inconsistency, indirectness, imprecision, and evidence of publication bias to assign a level of certainty to the body of evidence regarding each outcome or

comparison. Because this analysis was restricted to RCTs, we began with an assumption of high certainty evidence. Evidence was downgraded if a majority of studies in a given comparison were considered to be at unclear or high risk of bias. Inconsistency was assessed using $I^2$ values, with downgrading of one level for comparisons with an $I^2$ greater than or equal to 60%. To determine indirectness, factors such as population, the interventions and co-interventions, as well as the DI modality were considered. As the primary objective was to study the effect of AF on all diagnostic imaging utilization (X-ray, ultrasound, echocardiogram, CT, MRI), evidence was downgraded for indirectness if a comparison included two or fewer DI modalities. There is relatively little guidance in the literature on how to assess imprecision in reviews that use SMD as an outcome measure and when there is no clear consensus on what is considered a minimally important difference. We elected to follow the convention that a standardized mean difference (SMD) between 0.5 and 0.8 indicates a moderately effective intervention. If a CI was greater than the midpoint of that range, 0.65 units, we downgraded the certainty of evidence by one level [15]. For context, an SMD of 0.65 units in our primary analysis translates into a 14.3% reduction in DI ordering when converted into natural units as described above. Finally, to assess publication bias, we subjectively assessed the symmetry of the funnel plots (S3 and S4 Figs in S1 Appendix) and elected to downgrade evidence if there was clear asymmetry.

## Results

Eleven RCTs met the inclusion criteria from an initial literature search that identified 4493 papers (Fig 1) [27–37]. One non-randomized controlled trial (NRCT) [38] and 5 observational studies were included in the Appendix (S1-S4 Tables and S1, S2 Figs in S1 Appendix) [39–44]. All of these studies included at least one comparison that met our inclusion criteria. Some of the studies examined additional interventions which did not meet our inclusion criteria, and comparisons involving these other interventions were excluded. While the certainty of evidence for most comparisons was judged to be very low to low based on risk of bias, indirectness and imprecision (Table 1), we considered the evidence for the effect of AF on our primary outcome of all DI orders (both subgroups) to be of moderate certainty. The rationale for these certainty of evidence ratings is provided in the footnotes of Table 1 and the included studies are described in Table 2.

### Intervention fidelity, bias, and certainty of evidence

The AF interventions are described in Table 3. One study directly assessed if AF reports had been opened by tracking logins to an online system and found that 61% of participants logged in at least once [28]. Verstappen et al. reported that 100% of study participants attended an in-person education session at which AF reports were discussed [35] and O'Connor et al. reported the percentage of AF reports sent by post that were returned unopened was 4.9%-14.7%, dependent on which intervention they received [32]. However, the lack of a returned envelope was not considered sufficient proof of an AF receipt so we indicated "Not reported" for this variable. No other studies clearly reported AF receipt or other measures of intervention fidelity (Table 3).

The risk of bias was judged to be unclear in a majority of studies and only two studies were thought to be at low risk of bias (Table 4) [32, 35]. Nine of the eleven studies did not report on the concealment of study group allocation, but almost all studies were downgraded based on more than just that item. Zafar et al. [37] used hierarchical regression to adjust for clustering in some of their analyses; however, those results were not suitable for this meta-analysis. The data that are suitable for synthesis from this study do not appear to be adjusted for correlated

## Study identification, screening and exclusions

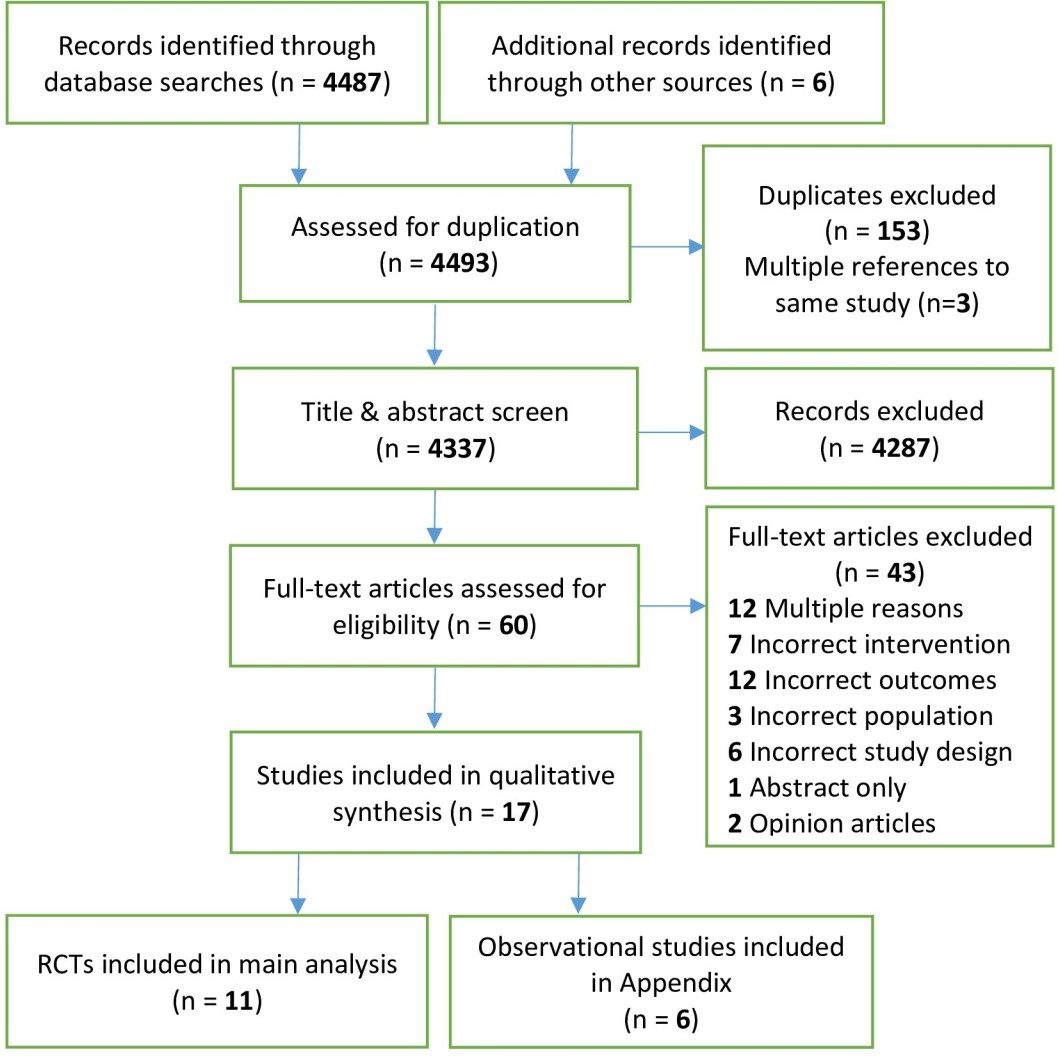

**Fig 1. PRISMA diagram for study identification, screening, and exclusions.**

observations and it was therefore reported as high risk for this reason. A risk of bias table was not included for the secondary outcome as evaluations were similar.

### The effect of AF on the total number of DI requests (Fig 2)

Ten trials examined the effect of AF on the number of diagnostic imaging requests, nine of which are presented in the forest plots (Fig 2). Six trials used a control group that did not receive any intervention or only received practice guidelines [27, 28, 30, 31, 32, 35] and three additional studies measured the effect of adding AF to another intervention [29, 33, 37]. The remaining study describes baseline imbalances between the control and intervention groups which results in very similar post-intervention outcomes (not shown) [36]. However, the authors of this study report a 4% reduction in test ordering over the study period in the

**Table 1. Summary of findings.**

**Effect of audit and feedback on diagnostic imaging requests**

**Population:** Healthcare providers
**Setting:** Healthcare
**Outcome:** Number of diagnostic imaging requests

| Comparison | Participants (studies) | Anticipated effects[a] | | Quality | Comments |
|---|---|---|---|---|---|
| | | Comparator (tests/1000 patients) | Intervention vs comparator MD (95% CI) | | |
| AF alone vs usual practice or guideline provision | 4190 | 31.5 tests | 1.4 fewer | ⊕⊕◯◯ | Negative MD favors AF |
| | (6 RCTs) | | (-2.8 to 0.0) | Low[c,d] | |
| AF+ other intervention vs other intervention alone | 107 | 20.3 tests | 1.7 fewer | ⊕◯◯◯ | 1 RCT examined CTPA and the other TTE |
| | (2 RCTs) | | (-4.3 to 0.9) | Very low [c,e,f] | |
| AF vs comparator (both subgroups) | 4297 | 31.3 tests | 1.5 fewer | ⊕⊕⊕◯ | |
| | (8 RCTs) | | (-2.6 to -0.4) | Moderate[c] | |

**Outcome:** Appropriateness of diagnostic imaging requests

| Comparison | Participants (studies) | Anticipated effects[a] | | Quality | Comments |
|---|---|---|---|---|---|
| | | Comparator (percent appropriate)[b] | Intervention vs comparator MD (95% CI) | | |
| AF alone vs usual practice or guideline provision | 317 | 82.9% | 0.9% greater | ⊕◯◯◯ | Positive MD favors AF |
| | (4 RCTS) | | (-4.5 to 6.3%) | Very Low[c,d,e,g] | |
| AF+ other intervention vs other intervention alone | 107 | - | 7.0% greater | ⊕◯◯◯ | 1 RCT examined CTPA and the other TTE |
| | (2 RCTs) | | (2.6 to 11.5%) | Very low[c,e,f] | |
| AF vs comparator (both subgroups) | 424 | 82.9% | 3.1% greater | ⊕◯◯◯ | |
| | (6 RCTs) | | (-1.5 to 7.7%) | Very low[c,d,e] | |

Abbreviations: AF: audit and feedback; CTPA: computed tomography pulmonary angiogram; MD: mean difference; RCT: randomized controlled trial; TTE: transthoracic echocardiogram

[a]. Note that a reduction in the number of orders and an increase in the appropriateness of orders were considered favorable outcomes. Thus, the sign of a favorable result is reversed for each outcome.

[b]. No papers in the second subgroup used the % appropriate outcome

GRADE rating explanations

[c]. Downgraded due to high or unclear risk of bias in a majority of studies. See Table 4. Most studies were unclear on their procedure for allocation concealment.

[d]. Downgraded due to inconsistency ($I^2 \geq 60\%$)

[e]. Downgraded due to imprecision. (95% CI for the SMD > 0.65)

[f]. Downgraded due to indirectness as studies only evaluated the effect of AF on CT scans for pulmonary embolism and transthoracic echocardiography.

[g]. Downgraded due to indirectness as 2 of the 4 studies only evaluated lumbar and knee radiographs, and the remaining 2 studies analyzed echocardiogram ordering

intervention group (p-value = 0.11), but they don't report control group data and we were therefore not able to include these results in the meta-analysis. The post-intervention values were not included in the meta-analysis because of bias due to the baseline imbalance [36]. Although there was a fair degree of heterogeneity in the types of imaging that were addressed by the AF interventions (Table 3), results for the pooled analyses of the primary outcome were only moderately heterogeneous ($I^2$ = 45%, p = 0.08) and heterogeneity existed mostly within potential subgroups (e.g. Bhatia 2014, 2017 and Dudzinski, 2016, all of which examined echocardiography), rather than between subgroups. We therefore decided not to pool our results by imaging modality and/or target organ.

The meta-analysis demonstrates a statistically significant reduction in total DI test ordering (SMD = -0.22, 95% CI = -0.38 to -0.06, p-value = 0.009), which translates into 1.5 fewer image test orders per 1000 patients seen (95% CI -2.6 to -0.4) in the intervention vs the control

**Table 2. Summary characteristics of included RCTs.**

| REFERENCE / COUNTRY | TARGET PROVIDER | TARGET BEHAVIOUR | AF DESCRIPTION | COMPARISON DESCRIPTION | SAMPLE SIZE (A) | PHYSICIAN/ PATIENT GENDER AND AGE | PRIMARY OUTCOME(S) | OUTCOME (B) |
|---|---|---|---|---|---|---|---|---|
| WINKENS, 1995 NL | Primary care physicians (c) | Decrease in various x-rays (d) and US orders | AF with critical feedback comments on individual test orders | Same intervention on different, non-imaging tests (No intervention) | 79 physicians (C:39, I:40) /~187 000 patients | Physicians: Proportion female (C: 10.2%, I: 10%) Patient: not reported | Mean number of tests ordered Mean percentage of tests which were guideline-appropriate | Total imaging Appropriateness |
| KERRY, 2000 UK | Primary care physicians | Decrease in all x-rays | AF alone | Paper guidelines | 69 practices (C:36, I:33)/ 175 physicians | Not reported | Mean percent reduction in total number tests ordered compared to baseline | Total imaging |
| ECCLES, 2001 UK | Primary care physicians | Decrease in lumbar spine and knee x-rays | AF alone | Paper guidelines | 121 practices (C:61, I:60) | Not reported | Mean number of tests/1000 patients Odds Ratio for an appropriate test in the intervention vs control groups | Total imaging Appropriateness |
| ROBLING, 2002 UK | Primary care physicians | Increase in guideline appropriate lumbar spine and knee MRIs | AF alone | Paper guidelines | 19 practices (C:10, I:9) 95 requests (C:53, I:42) | Not reported | Percentage of requests that were guideline appropriate | Appropriateness |
| VERSTAPPEN, 2003 NL | Primary care physicians | Decrease in x-rays of shoulder, spine, hip and knee | AF + clinician education | Same intervention for other (non-imaging) tests | 25 groups (C:12, I:13)/ 163 physicians (C:75, I:88) | Physicians: Proportion female (C: 16% control, I: 17% intervention). Mean age (C: 46%, I: 45.8%) Patients: Percent of patients over 65: (C: 15%, I: 13%) | Mean number of tests/ physician Mean number of guideline appropriate tests/physician | Total imaging Appropriateness |
| BHATIA, 2014 USA | Cardiology fellows | Decrease in rarely appropriate TTE | AF + clinician education + pocket card | No intervention | 1 hospital/24 physicians (C:12, I:12)/ 1213 patients (C:600, I:613) | Physicians: not reported Patients: Average age (C:65, I: 64) Proportion female (C: 32%, I: 33%) | Mean number of tests/ physician Proportion of tests that were guideline appropriate | Total imaging Appropriateness |
| RAJA, 2015 USA | Emergency physicians | Increase in guideline-appropriate and decrease in overall use of CTPA | AF + computerized decision support | Computerized decision support | 1 ED/43 physicians (C:21, I:22)/ 2167 patients (C:1149, I:1018) | Physicians: Mean age (C: 41.2, I: 39.4) Proportion female: (C: 29%, I:32) Patients: not reported. | Number of tests/patient seen Proportion of tests that were guideline appropriate | Total imaging Appropriateness |

(*Continued*)

**Table 2.** (Continued)

| REFERENCE / COUNTRY | TARGET PROVIDER | TARGET BEHAVIOUR | AF DESCRIPTION | COMPARISON DESCRIPTION | SAMPLE SIZE (A) | PHYSICIAN/ PATIENT GENDER AND AGE | PRIMARY OUTCOME(S) | OUTCOME (B) |
|---|---|---|---|---|---|---|---|---|
| DUDZINSKI, 2016 USA | Academic cardiologists | Decrease in rarely appropriate TTE | AF + clinician education | Clinician education | 1 hospital/ 66 physicians (C:33, I:33)/ 16075 tests (C:8166, I:7909) | Physicians: Age and gender not reported. Patients: Mean age (C: 66, I: 65) Proportion female (C: 39%, I: 41.8%) | Total tests ordered Proportion of tests that were guideline appropriate | Total imaging Appropriateness |
| BHATIA, 2017 CA, USA | Primary care physicians and cardiologists | Decrease in rarely appropriate TTE | AF + education | No intervention | 8 hospitals/ 153 physicians (C:79, I:74)/ 14697 tests (C:7798, I:6899) | Not reported | Mean number of tests/ physician Mean percentage of tests that were guideline-appropriate | Total imaging Appropriateness |
| ZAFAR, 2019 USA | Primary Care Physicians and Nurse practitioners | Decrease in LS MRI orders | AF + clinical decision support | Clinical decision support | 8 Practices/ 52 providers (C:26, I:26))/ 5142 visits (C:2021, I:3121) | Number not reported | Proportion of visits for LBP on which a test was ordered | Total imaging |
| O'CONNOR, 2022 AU | Primary Care physicians | Decrease in CT, MRI, X-Ray, US orders | AF alone | No intervention | 2271 practices/ 3660 physicians (C:727, I:2933) | Physicians: Proportion 60 or over (C: 42%, I: 40%) Proportion female (C: 37%, I: 39.5%) Patients: Not provided. | Rate of request/ 1000 patients | Total imaging |

[a]Data from this column that are missing were not reported. (I) refers to intervention; (C) refers to control. The highest level that includes numbers for (I) and (C) is the level at which experimental groups were randomized.

[b]Outcomes included total imaging or appropriate test orders as described in Methods

[c]Primary care physicians may include family, general practice and general internal medicine physicians

[d]Includes chest, cervical spine, thoracic spine, lumbar spine, pelvis/hip, knee, ankle and sinus x-rays

Abbreviations: AF, audit and feedback; AU, Australia; CA, Canada; CT, Computed Tomography; CTPA, CT Pulmonary Angiogram; LBP, low back pain; MRI, Magnetic Resonance Imaging; NL, Netherlands; TTE, Transthoracic Echocardiogram; US, Ultrasound; UK, United Kingdom; USA, United States of America.

groups. The GRADE quality of evidence for this summary effect was rated as moderate, but the rating for each subgroup was very low to low (Table 1). The weighted mean average number of DI tests ordered during the pre-intervention period of the three studies that used this outcome was 31.3 orders per 1000 patients seen [30, 32, 33]. Thus, audit and feedback was associated with a 4.9% (95% CI 1.3 to 8.4) greater reduction in test ordering than control. This finding is driven primarily by a single study which includes almost 70% of the participants in the meta-analysis [32]; however, the results of most other studies were similar ($I^2$ = 45%, p-value = 0.08). Only one study, which examined the effect of AF on echocardiogram ordering practices showed a higher rate of ordering in the AF vs the usual practice group, though this difference was not significant [27]. Interestingly, two other studies on echocardiogram

**Table 3. Description of AF interventions according to TiDIER recommendations.**

| | Winkens et al., 1995 | Kerry et al., 2000 | Eccles et al., 2001 | Robling et al., 2002 | Verstappen, et al. 2003 | Bhatia et al., 2014 | Raja et al., 2015 | Dudzinski et al., 2016 | Bhatia et al., 2017 | Zafar et al., 2019 | O'Connor et al 2020 |
|---|---|---|---|---|---|---|---|---|---|---|---|
| **Who and where?** | | | | | | | | | | | |
| Provider type | PCPs (a) | PCPs | PCPs | PCPs | PCPs | Card, GIM res. | EP | Cardio | PCPs, Card | PCPs | PCPs |
| AF provided to Individuals or group | Individual | Individual | Individual | Individual | Individual | Individual | Individual | Individual | Individual | Individual | Individual |
| AF delivered directly to provider? | Yes | Yes | Yes | Yes | Yes | Yes | Yes | Yes | Yes | Yes | Yes |
| Inpatient/ Outpatient setting? | Unclear | Unclear | Outpatient | Unclear | Outpatient | Outpatient | Inpatient | Outpatient | Outpatient | Outpatient | Outpatient |
| **Content of AF reports** | | | | | | | | | | | |
| Imaging modality | X-Ray, US | X-Ray | X-Ray | MRI | X-Ray | Echo | CTPA | Echo | Echo | MRI | CT, MRI, X-Ray, US |
| Desired change in ordering | Decrease | Decrease | Decrease | Decrease | Decrease | Decrease | Decrease | Decrease | Decrease | Decrease | Decrease |
| Patient outcomes (findings on imaging test) | Yes | No | No | No | No | Yes | No | No | No | No | No |
| Other info (e.g. costs, guidelines, doses) | Yes | Yes | No | No | Yes | Yes | No | No | Yes | No | Yes |
| AF of Individual or group | Individual | Group | Individual | Group | Individual | Individual | Individual | Individual | Both | Individual | Individual |
| Feedback about Individual cases or aggregate cases | Individual | Aggregate | Aggregate | Aggregate | Aggregate | Aggregate | Both | Aggregate | Aggregate | Aggregate | Aggregate |
| Comparison provided(b) | Own, Peers | Own | Peers | Peers | Peers | None | Peers | None | Own, peers | Own, peers | Peers |
| Explicit target provided | No | No | No | No | Yes | No | No | No | No | No | No |
| Action plan provided (c) | Yes | No | No | No | Yes | No | No | No | No | No | No |
| Graphical elements | No | No | No | No | Yes | No | Yes | No | Yes | No | Yes |
| **When and how much AF?** | | | | | | | | | | | |
| Time period of audit data | 1 Month | 6 Months | 6 Months | Unclear | 6 Months | 1 Month | 3 Months | 1 Month | 1 Month | Unclear | 12 months |
| Lag between audit and feedback | Weeks | Months | Months | Unclear | Unclear | Brief | Brief | Brief | Weeks | Months | Months |
| Frequency (number of times given) | 5 | 1 | 2 | 1 | 3 | 9 | 4 | 6 | 17 | Unclear | 1 or 2 |
| Time in between reports | 6–7 Months | N/A | 6 Months | N/A | Unclear | 1 Month | 3 Months | 1 Month | 1 Month | 4–6 Months | 12 months |

(Continued)

**Table 3.** (Continued)

| | Winkens et al., 1995 | Kerry et al., 2000 | Eccles et al., 2001 | Robling et al., 2002 | Verstappen, et al. 2003 | Bhatia et al., 2014 | Raja et al., 2015 | Dudzinski et al., 2016 | Bhatia et al., 2017 | Zafar et al., 2019 | O'Connor et al, 2020 |
|---|---|---|---|---|---|---|---|---|---|---|---|
| Duration of intervention (months) | 32 | 12 | 12 | Unclear | 6 | 9 | 12 | 6.4 | 17 | 11 | 12 or 24 |
| **How was AF delivered?** | | | | | | | | | | | |
| Verbal or written | Written | Written | Written | Both | Both | Written | Written | Written | Written | Written | Written |
| Delivery mode | Unclear | Unclear | Post | Post, In Person | Post, In Person | E-mail | E-mail | E-mail | E-mail | Unclear | Post |
| Source of in-person delivery | N/A | N/A | N/A | Expert | Leader | N/A | N/A | N/A | N/A | N/A | N/A |
| Asked to reflect on AF | Yes | No | No | Yes | Yes | No | No | No | No | No | No |
| **Who provided AF?** | | | | | | | | | | | |
| Who conducted the audit? | Authority (d) | Unclear | Researcher | Unclear | Unclear | Researcher | Researcher | Researcher | Researcher | Unclear | Researcher |
| Who developed the feedback report? | Authority (d) | Unclear | Researcher | Unclear | Unclear | Researcher | Researcher | Researcher | Researcher | Unclear | Unclear |
| **Fidelity** | | | | | | | | | | | |
| Planned assessment of AF receipt? | Not Reported | Not Reported | Not Reported | Not Reported | Yes | Not Reported | Not Reported | Not Reported | Yes | Not Reported | Not reported |
| AF receipt by providers | Not Reported | Not Reported | Not Reported | Not Reported | 100% | Not Reported | Not Reported | Not Reported | 61% | Not Reported | Not reported |

Abbreviations: AF, Audit and Feedback; Card, Cardiologists; CTPA, Computed Tomography Pulmonary Angiogram; Echo, Echocardiography; EP, Emergency Physicians; GIM, General physicians; MRI, Magnetic Resonance Imaging; N/A, not applicable; PCP, Primary care; Res, residents; US, ultrasound; XR, X-ray

(a) Primary care physicians may include family, general practice and general internal medicine physicians

(b) Comparison provided Includes own/ peers' previous performance, national benchmark

(c) Action plan refers to an explicit verbal or written plan to achieve desired targets. We did not consider the provision of appropriate use criteria separate from the AF reports to be an action plan.

(d) Refers to hospital, health maintenance organization (HMO) or health authority

Note: For multifaceted interventions, we assessed the characteristics of the audit and feedback component

ordering from the same research group showed the opposite, non-significant trend towards reduced ordering in the AF group [28, 29].

In the first subgroup of Fig 2A, Kerry et al., Eccles et al., and O'Connor et al. examined AF alone [30–32]. The remaining studies in this subgroup examined AF as the core part of a multi-faceted intervention, including a discussion or education session [27, 35], or an education session together with the provision of a mobile application to assist with decision-making [28]. In the second subgroup included in Fig 2A, the studies examined AF added to electronic clinical decision support [33] or an educational session [29]. The results of the two subgroups in this analysis were similar ($I^2$ = 0%, p-value = 0.83) suggesting that the effect of AF is similar when implemented on its own or when added to another intervention, although only 2 studies were included in the second subgroup. The additional study included in Fig 2B (dichotomous outcome), which investigated the effect of AF added to real-time alerts implemented at the point of electronic ordering [37], reports similar findings.

**Table 4. Risk of bias for each domain and overall judgement for risk of bias for each included RCT.**

| Bias | Winkens 1995 | Kerry 2000 | Eccles 2001 | Robling 2002 | Verstappen 2003 | Bhatia 2014 | Raja 2015 | Dudzinski 2016 | Bhatia 2017 | Zafar 2019 | O'Connor 2022 |
|---|---|---|---|---|---|---|---|---|---|---|---|
| Random Sequence Gen. | Unclear | Low | Low | Unclear | Low | Low | Low | Low | Unclear | Unclear | Low |
| Allocation concealment | Unclear | Unclear | Unclear | Unclear | Low | Unclear | Unclear | Unclear | Unclear | Unclear | Low |
| Blinding of participants | Low | Low | Low | Low | Low | Low | Low | Low | Low | Low | Low |
| Blinding of outcome assessment | High | Low | Low | Low | Low | Low | Low | Low | Low | Low | Low |
| Incomplete outcome data | Low | Low | Low | High | Low | Low | Low | Low | Low | High | Low |
| Selective reporting | Low | Low | Low | Low | Low | Low | Low | Low | Low | Low | Low |
| Other (baseline imbalance) | Unclear | Unclear | High | Unclear | Low | High | Low | Low | Low | Unclear | Low |
| Other (no clustering adjustment) | Low | Low | Low | Low | Low | Low | Unclear | Low | Low | High | Low |
| | | | | | | | | | | | |
| OVERALL JUDGMENT | High | Unclear | High | High | Low | High | Unclear | Unclear | Unclear | High | Low |

● Low risk; ● Unclear Risk; ● High risk

## The effect of AF on the appropriateness of diagnostic imaging requests (Fig 3)

Whereas a decrease in total imaging was considered favorable, an increase in appropriateness was considered favorable. Thus, studies favoring AF are presented on opposite sides of the vertical axis in the forest plots for each of these outcomes (Figs 2 and 3). Four studies evaluated the effect of AF on the appropriateness of DI requests compared to usual practice, [27, 28, 30, 34] and two additional studies evaluated AF added to electronic clinical decision support [33] or an educational session [29]. All studies included in this section were also included in the primary outcome analyses (Fig 2A), with the exception of Robling et al. [34]. Results for the appropriateness outcome were mixed compared to the total imaging outcome, but overall, AF had no significant effect on appropriateness (SMD = 0.27, 95% CI = -0.13, 0.66, p-value = 0.18), with a high degree of heterogeneity ($I^2$ = 70%, p-value = 0.005). This SMD translates into a 3.1% (95% CI -1.5 to 7.7%) higher proportion of image orders that were considered to be appropriate in the AF vs the comparator groups. Three of the six studies that examined appropriateness were deemed to be at high risk and the remainder were deemed to be at unclear risk of bias.

The two studies that examined AF added to another intervention were consistent ($I^2$ = 0%, p-value = 0.48) in finding that AF improved appropriateness (SMD = 0.60, 95% CI = 0.22, 0.99, p-value = 0.002), despite substantial differences in the co-interventions examined in those two studies.

## The effect of AF alone vs the effect of AF added to another intervention (subgroup 1 vs subgroup 2)

The effect of AF alone is presented in subgroup 1 and the effect of AF added to another intervention is presented in subgroup 2 of Figs 2 and 3. Although one might expect that co-interventions would "dilute" the effectiveness of AF, our findings suggest that may not be the case,

**Fig 2. Effect of audit and feedback on the number of diagnostic imaging requests.** The AF groups in this figure include audit and feedback alone and audit and feedback as the main component of a multi-faceted intervention. The control group includes usual care or the provision of paper guidelines only (subgroup 1) or an active control group that was compared against the same intervention with the addition of AF (subgroup 2). Note that the results from Raja et al and Zafar et al may not be adjusted for correlated observations.

albeit on the basis of a limited number of studies. The SMD for AF added to another intervention for both outcomes is higher than the SMD for AF alone, although this is only significant for the appropriateness outcome.

## Results from observational studies (S1 Appendix)

The meta-analyses of the observational study data were similar to those included in the main text, with a higher degree of variability contributing to non-significant summary results. Almost all studies were considered to be at high risk of bias (S2 Table in S1 Appendix). The single study judged to be at low risk of bias was an interrupted time series analysis of clinical data related to a national intervention to improve the management of back pain, including a reduction in the use of imaging tests [43]. This study is notable because of its strong design that mitigates many of the limitations of observational analyses, its low risk of bias and the dramatic 10.9% reduction in imaging (albeit with a high degree of imprecision: 95% posterior interval = 0.85–20.9%) after the introduction of their AF intervention, resulting in substantial cost savings [43].

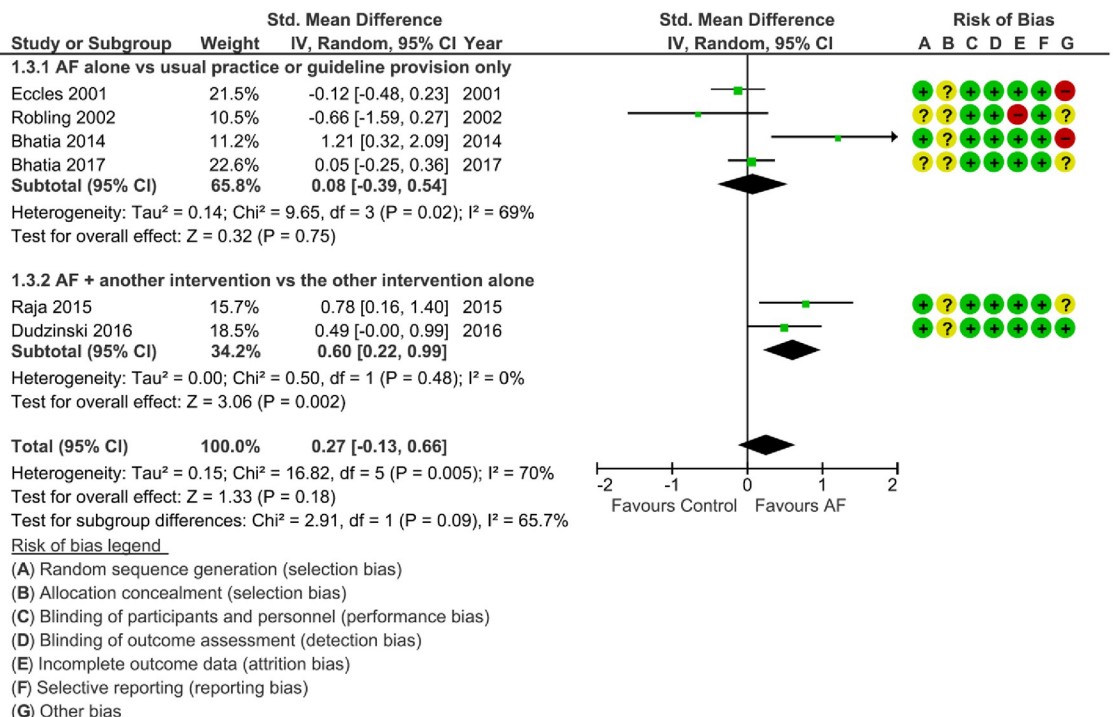

**Fig 3. Effect of audit and feedback on the appropriateness of diagnostic imaging requests.** The AF groups in this figure include audit and feedback alone and audit and feedback as the main component of a multi-faceted intervention. The control group includes usual care or the provision of paper guidelines only (subgroup 1) or an active control group that was compared against the same intervention with the addition of AF (subgroup 2). Although Dudzinski et al and Bhatia et al (2017) papers found no significant difference in the appropriateness outcome analyzed in our meta-analysis, both papers found a significant reduction in "rarely appropriate" echocardiograms in their AF intervention group (Odds Ratio (OR) = 0.59, 95% CI 0.39–0.88, p = 0.01 and OR = 0.75, 95% CI 0.57–0.99, p = 0.039, respectively). Note that favors AF is on the right side of the axis.

## Discussion

This review includes 11 RCTs that assessed the effect of audit and feedback on diagnostic image test ordering. Our meta-analyses demonstrated a significant, 4.9% reduction in total number of DI orders but variable and non-significant results on the appropriateness of orders. The evidence for the primary, total DI orders outcome was judged to be of moderate certainty but the evidence for all other comparisons was found to be very low to low certainty and these final results should therefore be interpreted with caution. For context, the Cochrane review on all uses of AF for healthcare found a roughly 1.3% improvement in practice associated with AF interventions [11]; thus, the effectiveness of AF appears to be larger when used on DI ordering. The two studies that were deemed to be at low risk of bias in our review contrasted in their findings, with one study finding a significant, modest reduction in DI ordering after AF [32], while the other found no significant effect [35]. Neither of these low-risk studies examined the appropriateness of DI requests; thus, the results for this outcome must be interpreted with greater caution.

All studies that reported appropriateness expressed this outcome as a proportion of total image orders. Our finding that AF interventions result in a decrease in total image ordering but no statistically significant change in the proportion of appropriate orders, suggests that appropriate and inappropriate tests may therefore be reduced at a similar rate following AF. This may increase the risk of delayed or missed diagnoses due to a reduction in appropriate

testing and disproportionately harm people who generally receive lower imaging rates, particularly minority groups and people of color[45, 46]. However, DI appropriateness criteria are relatively crude measures that often do not address a substantial grey area in clinical decision-making; thus, we cannot infer that reductions in "appropriate" imaging automatically result in patient harm [27, 29].

Although our meta-analyses demonstrated no significant effect on the appropriateness outcome, several papers found a significant benefit on a related outcome. While the effect of AF on appropriateness in Dudzinski et al. and Bhatia et al. 2017 [28, 29] was non-significant (Fig 3), these authors found significant effects on "rarely appropriate" (i.e., inappropriate) imaging requests. Bhatia et al. 2014 [27] found significant effects on both appropriate and rarely appropriate imaging requests. This discordance in the statistical significance of two related outcomes (appropriateness and inappropriateness) is not unexpected, especially when there is a substantial difference in the frequency of these outcomes. The work of the Cochrane collaboration demonstrates that statistical significance is more likely for less frequent outcomes [15, Section 6.4.1.5]. We chose to analyze "appropriateness" rather than "inappropriateness," as not all papers reported both outcomes and this allowed a greater number of studies to be included in our meta-analyses.

### Recommendations to enhance the effectiveness of AF

A meta-regression completed as part of the Cochrane AF review found that low baseline performance, repeated delivery of AF reports, a supervisor or colleague as the source of feedback, both verbal and written delivery of feedback and the provision of explicit targets and an action plan were all associated with improved effectiveness of AF [11]. While most of the studies included in our review did not comment on baseline performance, the single study with the most dramatic effect on DI ordering selectively enrolled high test-ordering clinicians [32]. This study also found that receiving two instances of AF reduced ordering to a greater degree than one report [32]. The frequency of AF provision amongst the other studies included in our review range from one to seventeen. Comparing across these studies, we did not observe an association between the numbers of reports received and reduced ordering; in fact, the largest effect sizes were observed in the studies that provided one to two reports. Our review does not support the recommendations that AF is provided by a supervisor or colleague, that AF should be provided both verbally and written, or that specific targets or action plans be provided with AF, albeit on the basis of a limited number of studies that examined these aspects. Thus, our results should be considered inconclusive regarding the effectiveness of these features in AF for DI requests.

Although we did not find support for the recommendation that AF reports be delivered by a supervisor or colleague, presumably the value of this method is the perception of reliability and importance of the information. This factor is often considered critical when pursuing clinician behaviour change [47, 48]. Additionally, having reports delivered by a supervisor may motivate clinicians to change their behavior to maintain their professional reputation with their supervisors and peers [49]. In the three studies focusing on echocardiogram ordering, the greatest benefit came in the study targeting cardiology and general internal medicine residents compared to other studies that enrolled independently practicing physicians [27–29]. While these 3 studies did not include delivery of AF reports by an individual, it may be that the clinicians in training were more likely to perceive the information as trustworthy or they were more motivated by a desire to achieve professional norms [48].

Another factor that was not examined in the Cochrane meta-regression [11] was the effect of visual appearance on the effectiveness of AF. While the four studies in our review that

included graphical elements in their AF reports do not appear to be associated with improved AF performance, O'Connor et al. compared an enhanced to a standard visual display of their AF data in their factorial design trial, and found that the enhanced display outperformed the standard version [32]. In this study, both standard and enhanced versions of the report included graphical information, but the enhanced version added highlighting to draw attention to indicators of higher utilization. Enhanced visual displays such as this could increase the effectiveness of AF, without substantially increasing costs and resource utilization.

A final consideration is the effectiveness of AF alone verses the effect when AF is added to another intervention. The evidence is very low certainty, but our findings suggest the possibility that AF may be more effective when added to another intervention than it is when implemented alone.

## Limitations

Most of the studies included in this review were determined to be either at high or unclear risk of bias, and the quality of evidence for most comparisons was assessed as very low to low. Although the papers included in this review examined a range of imaging modalities, six of the studies exclusively examined a single, less commonly used imaging modality, sometimes just for a specific indication such as pulmonary embolism, or knee and back pain. The effectiveness of AF may vary across different modalities or indications and therefore these findings may be hard to generalize across different modalities and indications. While all the imaging modalities are used for diagnosis, some of the tests such as echocardiogram are more commonly used to monitor for progression of previously diagnosed conditions such as valvular heart disease, congestive heart failure and aortic dilation than they are for the initial diagnosis of those conditions, which may also affect the results of an AF intervention. Because the effectiveness of AF may vary dependant on indication or imaging modality, future studies could restrict the analyses to further investigate the effect of AF on these specific indications or modalities.

## Conclusions

This review reports moderate quality evidence that AF and AF added to other interventions likely has a small but variable effect on the total number of DI requests, but results for improvements in the appropriateness of those requests are equivocal and of very low quality. The observation that AF may reduce total imaging requests with no change in appropriateness, suggests that both clinically indicated and inappropriate tests are reduced at a similar rate, raising the possibility of adverse clinical outcomes. Future studies of AF interventions should pay careful attention to study design and reporting standards to improve the quality and reliability of evidence, and they should consider studying harm outcomes.

## Supporting information

**S1 Appendix. S1 Fig.** a. Effect of audit and feedback in observational studies on the number of diagnostic imaging requests (continuous outcome) (4–6). b. Effect of audit and feedback in observational studies on the number of diagnostic imaging requests (dichotomous outcome) (7, 8). **S2 Fig. Effect of audit and feedback in observational studies on image order appropriateness (dichotomous outcome) (7). S3 Fig. Funnel plot of RCTs analyzing the total image order outcome.** We did not consider this figure to be indicative of publication bias. The study in the bottom right favored the control intervention, not AF. **S4 Fig. Funnel plot of RCTS analyzing the appropriateness of image orders outcome.** We did not consider this figure to be indicative of publication bias. **S1 Table. Description of AF interventions using TiDIER recommendations (1).** Abbreviations: AF, Audit and Feedback; CT, Computed

Tomography; Echo, Echocardiography; GIM, General physicians; Res, residents; Gov., Government; Mm; MRI, Magnetic Resonance Imaging; N/A, not applicable; PCP, Primary care physicians (e) PCPs refers to primary care physicians and may include family, general practice and general internal medicine physicians, (f) The term residents also refers to registrars (g) Comparison provided Includes own/ peers' previous performance, national benchmark. Note: For multifaceted interventions, we assessed the characteristics of the audit and feedback component. **S2 Table.** a. Risk of Bias for NRCTs using the Risk Of Bias In Non-randomized Studies—of Interventions (ROBINS-I) tool (2). b. Risk of Bias for observational studies using Effective Practice and Organisation of Care (EPOC) recommendations (3). c. Risk of Bias for interrupted time series studies using Effective Practice and Organisation of Care (EPOC) recommendations (3). Legend: ● Low risk; ● Indeterminate Risk; ● High risk. **S3 Table. Effect of audit and feedback in a non-randomized, crossover design study on the number of diagnostic imaging request 9).**\*no p-values, standard deviations, or confidence intervals provided. **S4 Table. Effect of audit and feedback with another intervention vs usual care in an interrupted time-series analysis (10). S1 File. EMBASE search strategy. S2 File. CINAHL search strategy. S3 File. PubMED search strategy. S4 File. References for supporting information.** (ZIP)

**S1 Checklist. PRISMA checklist.**
(DOCX)

## Acknowledgments

We would like to acknowledge Bethan Copsey from the University of Leeds for her advice on the statistical considerations for the meta-analyses.

## Author Contributions

**Conceptualization:** Oluwatosin Badejo, Amanda Hall, Bradley Furlong, Zhiwei Gao, Kris Aubrey-Bassler.

**Formal analysis:** Oluwatosin Badejo.

**Investigation:** Maria Saleeb.

**Methodology:** Oluwatosin Badejo, Maria Saleeb, Amanda Hall, Zhiwei Gao, Brendan Barrett, Lindsay Alcock, Kris Aubrey-Bassler.

**Resources:** Lindsay Alcock.

**Supervision:** Amanda Hall, Brendan Barrett, Kris Aubrey-Bassler.

**Validation:** Lindsay Alcock.

**Writing – original draft:** Oluwatosin Badejo, Maria Saleeb, Kris Aubrey-Bassler.

**Writing – review & editing:** Oluwatosin Badejo, Maria Saleeb, Amanda Hall, Bradley Furlong, Gabrielle S. Logan, Zhiwei Gao, Brendan Barrett, Kris Aubrey-Bassler.

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
