## [Decision Letter · Decision Letter 0]

28 Nov 2023

PONE-D-23-34965Audit and Feedback to change diagnostic image ordering practices: A systematic review and meta-analysis.PLOS ONE

Dear Dr. Aubrey-Bassler,

Thank you for submitting your manuscript to PLOS ONE. After careful consideration, we feel that it has merit but does not fully meet PLOS ONE’s publication criteria as it currently stands. Therefore, we invite you to submit a revised version of the manuscript that addresses the points raised during the review process. Both reviewers have provided a comprehensive assessment of the paper and raised important points to consider.

We look forward to receiving your revised manuscript.

Kind regards,

Joshua Robert Zadro, PhD

Academic Editor

PLOS ONE

Journal Requirements:

Reviewers' comments:

Reviewer's Responses to Questions

**Comments to the Author**

1. Is the manuscript technically sound, and do the data support the conclusions?

Reviewer #1: Yes

Reviewer #2: Yes

2. Has the statistical analysis been performed appropriately and rigorously? 

Reviewer #1: No

Reviewer #2: I Don't Know

3. Have the authors made all data underlying the findings in their manuscript fully available?

Reviewer #1: Yes

Reviewer #2: Yes

4. Is the manuscript presented in an intelligible fashion and written in standard English?

Reviewer #1: Yes

Reviewer #2: Yes

5. Review Comments to the Author

Reviewer #1: PONE-D-23-34965

Audit and Feedback to change diagnostic image ordering practices: A systematic review and meta-analysis

Comments to editor

Thank you for the opportunity to review this manuscript.

Comments to author

I thank the authors for providing the opportunity to review this manuscript. The authors aimed to assess the effect of auditing and feedback interventions directed at healthcare workers to reduce imaging orders. The authors ran a comprehensive search and included 11 RCT’s with 4311 participants in total. This review provides an interesting insight into the effects of audit and feedback interventions on the ordering of images in primary care, especially given the high rates of unnecessary imaging currently. There are some matters that will need to be addressed before this review could be considered suitable for publication.

Primarily, the concerns are regarding the way in which the studies have been pooled together. Different conditions will have a lower or higher threshold for further testing and pooling these studies together may not appropriate. For example, studies done on cardiologist will yield very different results to studies in a musculoskeletal setting as one condition (chest pain) would be considered life threatening compared to the other (i.e., knee pain)

Major issues/comments

OVERALL

- My impression is that the authors have looked at the effect of AF intervention to reduce imaging. My concern is that the pooling of studies with different conditions (i.e., musculoskeletal based studies vs cardiovascular studies) includes too much heterogeneity and means the results are not entirely interpretable. This is because the threshold for serious pathology will be different e.g., chest pain will likely get imaged and have a low threshold for further testing, whereas knee pain may have a higher threshold for further testing as it may not be considered life threatening.

ABSTRACT

METHODS

- One overall comment is that it may not be appropriate to pool studies that used diagnostic imaging for different body parts. For example, the threshold for suspicion of serious pathology is going to be different for the lumbar spine verses a pulmonary condition. The setting of which the imaging is ordered is also something that should be pooled separately. Serious pathology is more prevalent in emergency departments settings and clinicians may have a lower threshold to image compared to primary care. Therefore, pooling all the studies together does not seem appropriate due to the heterogeneity between studies. My suggestion would be pool studies based on musculoskeletal conditions, respiratory, cardiovascular etc. Happy for this to be discussed with the Editor, however a meta-analysis inclusive of all body parts seems inappropriate in my view.

RESULTS

Study Characteristics

- No description of participant characteristics e.g., age, gender

Overall comment

- This section is written well on a complicated issue. However as per the comments in the methods section, I do not feel that the way the data is presented accurately highlights the impact of AF interventions, as the effect will differ for different body parts. This is evident for example in Figure 2A Eccles 2001 (musculoskeletal) and Bhatia 2014 (cardiology) and the varying results of these two studies.

DISCUSSION

Strengths and limitations

- If the Editor decides the current meta-analysis is ok, then I think the heterogeneity between the studies need to be clearly stated. It needs to be clear that the pooling of different conditions adds to the heterogeneity in the data and means the interpretation of these results should be taken very cautiously. Again, my recommendation is to reconsider the methods.

Reviewer #2: Thank you for the opportunity to review this manuscript titled “Audit and Feedback to change diagnostic image ordering practices: A systematic review and meta-analysis.”. This is an important topic and will be of interest to administrators and policy makers looking to reduce overuse of imaging which wastes resources and can worsen patient outcomes. This is a high quality systematic review that follows EPOC recommendations, uses the TIDieR statement describe interventions and was prospectively registered.

6. PLOS authors have the option to publish the peer review history of their article (what does this mean?). If published, this will include your full peer review and any attached files.

Reviewer #1: No

Reviewer #2: **Yes: **Gemma Altinger

---

## [Author Response · Author response to Decision Letter 0]

8 Feb 2024

Please see our response attached to this portal. We have also provided it here for convenience. 

February 7, 2024

Dear Editors,

We thank the reviewers for providing positive feedback and the opportunity to revise our manuscript. In response to the comments, suggestions, and guidance received from the reviewers, we have modified our manuscript and addressed the feedback. We hope our revisions will justify publication. Please note that we have grouped reviewer comments that are related and require a similar response, and we have not listed positive reviewer comments that did not require a response from us. Additionally, please note that references 5&6 have been added to our manuscript as they are more recent than the previous ones, and references 22 & 23 have been included to notify of the change from the Cochrane risk bias tool version 1 to the updated version 2. 

RESPONSE TO REVIEWER 1:

Comments to author 

-Primarily, the concerns are regarding the way in which the studies have been pooled together. Different conditions will have a lower or higher threshold for further testing and pooling these studies together may not appropriate. For example, studies done on cardiologist will yield very different results to studies in a musculoskeletal setting as one condition (chest pain) would be considered life threatening compared to the other (i.e., knee pain)

Comment 2: 

OVERALL

- My impression is that the authors have looked at the effect of AF intervention to reduce imaging. My concern is that the pooling of studies with different conditions (i.e., musculoskeletal based studies vs cardiovascular studies) includes too much heterogeneity and means the results are not entirely interpretable. This is because the threshold for serious pathology will be different e.g., chest pain will likely get imaged and have a low threshold for further testing, whereas knee pain may have a higher threshold for further testing as it may not be considered life threatening. 

Comment 3: 

METHODS

- One overall comment is that it may not be appropriate to pool studies that used diagnostic imaging for different body parts. For example, the threshold for suspicion of serious pathology is going to be different for the lumbar spine verses a pulmonary condition. The setting of which the imaging is ordered is also something that should be pooled separately. Serious pathology is more prevalent in emergency departments settings and clinicians may have a lower threshold to image compared to primary care. Therefore, pooling all the studies together does not seem appropriate due to the heterogeneity between studies. My suggestion would be pool studies based on musculoskeletal conditions, respiratory, cardiovascular etc. Happy for this to be discussed with the Editor, however a meta-analysis inclusive of all body parts seems inappropriate in my view. 

Comment 5: 

Overall comment

- However as per the comments in the methods section, I do not feel that the way the data is presented accurately highlights the impact of AF interventions, as the effect will differ for different body parts. This is evident for example in Figure 2A Eccles 2001 (musculoskeletal) and Bhatia 2014 (cardiology) and the varying results of these two studies.

Comment 6: 

DISCUSSION 

Strengths and limitations

- If the Editor decides the current meta-analysis is ok, then I think the heterogeneity between the studies need to be clearly stated. It needs to be clear that the pooling of different conditions adds to the heterogeneity in the data and means the interpretation of these results should be taken very cautiously. Again, my recommendation is to reconsider the methods. 

Response 1: 

We sincerely thank the reviewer for their time and effort reviewing our manuscript. We acknowledge the point raised about appropriate grouping of studies and we have put a lot of thought into this issue. Our results for the primary outcome of total DI utilization are relatively homogeneous (I2=45%, Figure 2A). The primary source of heterogeneity for the pooled analyses (Bhatia 2014) examines the use of echocardiography in cardiology learners and is very similar to two other papers that examine the use of echo, mostly in cardiologists (Dudzinski 2016 and Bhatia 2017), both of which had outcomes that were similar to the pooled result. i.e. the main heterogeneity lies within one of the proposed pools rather than between pools. Given this issue and the small number of studies included in this review, we worry that adding additional layers of pooling will complicate data interpretation and data presentation may suffer; we therefore propose to maintain the pooling as it currently stands. In support of this approach, the 2012 Ivers Cochrane review (1), the largest systematic review on the subject of audit and feedback (AF), grouped studies of healthcare AF interventions with substantially more variability than our own, ranging from chronic disease management to antibiotic prescribing and diagnostic imaging utilization.

We have added several lines summarizing the rationale provided here to our paper (Lines 213-217)

1. Ivers N, Jamtvedt G, Flottorp S, Young JM, Odgaard-Jensen J, French SD, et al. Audit and feedback: effects on professional practice and healthcare outcomes. Cochrane Database Syst Rev. 2012(6):Cd000259.

RESULTS

Comment 4: 

Study Characteristics

- No description of participant characteristics e.g., age, gender

Response 2: 

Thank you for bringing this to our attention. We have now included participant and patient age and gender when available in Table 2. 

RESPONSE TO REVIEWER 2: 

Reviewer #2

Comment 1: 

Introduction:

Well written, clear, and relevant.

Line 5 – I think it would be more powerful to mention the types of unnecessary treatments, e.g. surgery and opioid use in the context of back pain/ The treatment cascade can have significant risk of harm, particularly if it arose from unnecessary testing. This could be emphasised. 

Line 12 – Could you add the GRADE rating/certainty of evidence for the Cochrane review here?

Response 3: 

Thank you for giving us a chance to strengthen our introduction. From lines 4-9, the impact of unnecessary treatments has been further emphasized. The GRADE rating of moderate has been included for the Cochrane review on line 16. 

Comment 4: 

Methods:

Lines 84 to 87 – Could you please expand further on how you judged appropriateness, e.g. which guidelines you compared the practice to, to decide if it was appropriate. Or, did the review authors reply on what the trial authors deemed appropriate? This could be more clear.

Response 4: 

This has been clarified in the Methods on line 90 to explain that appropriateness was judged by the individual study authors. Thank you for bringing this to our attention. We did not have access to individual DI test ordering data that would have been necessary to gauge appropriateness ourselves. 

Comment 4: 

Line 109 – Please specify if you used the Cochrane ROB tool 1 or 2

Response 4: 

This has been specified on line 113. 

Comment 5: 

Lines 110 to 112 – Significant differences in base characteristics would be by nature due to random chance, assuming randomisation procedure was appropriate. Issues with randomisation would be picked up with the Cochrane ROB tool, therefore I don’t think this extra step is necessary. 

Response 5: 

Thank you for giving us the opportunity to explain how this downgrade was decided on. We downgraded the evidence in studies because of baseline imbalance that was “greater than expected due to chance.” The Cochrane collaboration allows for results to be downgraded for this reason (Cochrane Handbook, Section 8.3.2). 

One of the papers in our review reported a significant benefit of AF in reducing DI ordering in terms of a change from baseline measure (comment in text only - data not shown), but the post-intervention ordering data which was presented suggested that the effect of AF was essentially neutral (ordering in fact slightly increased in the AF group). We not only downgraded the risk of bias assessment for this paper, but we also decided to exclude this result from out meta-analysis, both of which were justified in our opinion. Excluding the result from the meta-analysis would have been harder to justify had we not downgraded the evidence for baseline imbalance. 

Comment 6: 

Line 115 – In similar reviews, lack of a pre-published protocol is considered a risk for selective reporting. Could you please elaborate on why you did not consider this to be a risk?

Response 6: 

Because our primary outcome was objective, we decided not to consider this a risk. We have added a comment on line 120 to clarify this.

Comment 7: 

Results:

Line 212 – Given the studies ranges from very low-moderate quality, consider clarifying that it is a low to moderate certainty evidence that AF can reduce imaging.

Response 7: 

This has been added on lines 221-223, thank you for this suggestion.

Comment 8:

Line 233 – word “to” not required

Response 8: 

This has been fixed, thank you for alerting us.

Comment 9: 

Discussion:

Line 280-287 - Another implication that could be considered is how AF may reduce imaging for those who are typically underserviced, increasing harm to these patients. For instance, interventions and recommendations to reduce antibiotics for all children with otitis media can harm Indigenous children who are underserviced and under treated. Are there certain populations that could experience fewer DI than is clinically indicated? 

Response 9: 

This is a very important point to consider, and thank you for bringing this to our attention. This has been added to the discussion (line 296), as it has been documented that people of colour receive less diagnostic imaging than their white counterparts, and could be disproportionately affected by a reduction of appropriate imaging tests. 

Comment 10: 

Line 291 – It is interesting that “rarely appropriate” imaging requests had significant effects. You have provided the explanation of frequency, but could there also be clinician agreement and endorsement of the inappropriateness of imaging in those cases which increased the effect?

Response 10

Thank you for allowing us to explain our rationale. Any reduction in inappropriateness should result in an increase in appropriateness (when expressed as proportions), unless providers only changed their documentation without any change to their ordering practices. However, if this was the case, we would expect total ordering (our primary outcome - figure 2) to be unchanged, which was not the case.

Alternatively the reviewer may have understood that appropriateness was determined in part by the ordering clinicians; however, that is not the case - in all studies, appropriateness was determined retrospectively by (usually) blinded study personnel reviewing ordering materials. Please feel free to have the reviewer clarify this comment so that we might address it better. 

Comment 11: 

Line 315 – Other reasons a supervisor of colleague may be used is because this can threaten professional status and peer opinion. Clinicians may be motivated by wanting to maintain their reputation.

Response 11: 

Thank you for this suggestion as this likely another motivating factor for physicians when being engaged in AF through their supervisor. This has been added to the discussion from lines 330-331. 

Comment 13: 

Tables and Figures:

Figure 1 and 2 have AF on the left, but figure 3 has AF on the right. At a glance, this looks like the effects were in the opposite direction. If possible, keep consistent across figures. 

Otherwise tables and figures are excellent.

Response 13: 

This is a great point as this may confuse readers. A favourable primary outcome was considered to be negative (reduction in total image ordering) whereas a favourable secondary outcome is positive (increase in appropriateness). We debated presenting the data as the reviewer suggests, but elected not to as that may cause confusion for other readers. This explanation has been added to the legend of Figure 3 and bolded to clear up any confusion for readers. 

Comment 14:

Please submit PRISMA checklist.

Response 14:

An older version of the PRISMA checklist was attached with our previous submission, but we have now included the most up to date version. Several changes were made to the abstract to meet the PRISMA abstract checklist and to keep the length less than 300 words.

Sincerely,

Kris

Kris Aubrey-Bassler

Associate Professor and Director

On behalf of the co-authors

---

## [Decision Letter · Decision Letter 1]

20 Feb 2024

Audit and Feedback to change diagnostic image ordering practices: A systematic review and meta-analysis.

PONE-D-23-34965R1

Dear Dr. Aubrey-Bassler,

We’re pleased to inform you that your manuscript has been judged scientifically suitable for publication and will be formally accepted for publication once it meets all outstanding technical requirements.

Kind regards,

Joshua Robert Zadro, PhD

Academic Editor

PLOS ONE

Additional Editor Comments (optional):

I thank the authors for adequately addressing the reviewers comments and congratulate them on putting together a very important paper.

Reviewers' comments:

Reviewer's Responses to Questions

**Comments to the Author**

1. If the authors have adequately addressed your comments raised in a previous round of review and you feel that this manuscript is now acceptable for publication, you may indicate that here to bypass the “Comments to the Author” section, enter your conflict of interest statement in the “Confidential to Editor” section, and submit your "Accept" recommendation.

Reviewer #2: All comments have been addressed

2. Is the manuscript technically sound, and do the data support the conclusions?

Reviewer #2: Yes

3. Has the statistical analysis been performed appropriately and rigorously? 

Reviewer #2: I Don't Know

4. Have the authors made all data underlying the findings in their manuscript fully available?

Reviewer #2: Yes

5. Is the manuscript presented in an intelligible fashion and written in standard English?

Reviewer #2: Yes

6. Review Comments to the Author

Reviewer #2: (No Response)

7. PLOS authors have the option to publish the peer review history of their article (what does this mean?). If published, this will include your full peer review and any attached files.

Reviewer #2: **Yes: **Gemma Altinger

---

## [Editor Report · Acceptance letter]

4 Mar 2024

PONE-D-23-34965R1 

PLOS ONE

Dear Dr. Aubrey-Bassler, 

I'm pleased to inform you that your manuscript has been deemed suitable for publication in PLOS ONE. Congratulations! Your manuscript is now being handed over to our production team.

Kind regards, 

on behalf of

Dr. Joshua Robert Zadro 

Academic Editor

PLOS ONE